# Non-Contact In-Plane Movement Estimation of Floating Covers Using Finite Element Formulation on Field-Scale DEM

**Leslie Wong** [1,*] , **Benjamin Steven Vien** [1] , **Thomas Kuen** [2] , **Dat Nha Bui** [1] , **Jayantha Kodikara** [3] **and Wing Kong Chiu** [1]

1   Department of Mechanical & Aerospace Engineering, Monash University, Clayton, VIC 3008, Australia
2   Melbourne Water Corporation, 990 La Trobe Street, Docklands, VIC 3008, Australia
3   Department of Civil Engineering, Monash University, Clayton, VIC 3008, Australia
*   Correspondence: leslie.wong@monash.edu

**Abstract:** Unmanned aerial vehicle (UAV) assisted photogrammetry has been used to perform a non-contact measurement of the covers on the anaerobic wastewater lagoons at Melbourne Water Corporation's Western Treatment Plant (WTP). These floating covers are valuable assets that eliminate odour and greenhouse gas emissions and harvest the methane-rich biogas as a renewable resource to generate electricity. Hence, the state of deformation of the floating covers becomes an important engineering factor for structural integrity assessment. UAVs have been deployed to scan these covers and photogrammetry has been used to process the aerial images to construct the floating covers' orthophoto and digital elevation model (DEM). This paper proposes to adopt the finite element formulation to improve the quantification of the in-plane movement of a floating cover. Distinguishable features on the floating cover are first identified and their ($x$, $y$ and $z$) coordinates are recorded over time. The results show that the technique can be used to quantify the short-term and long-term relative global lateral movement of the floating covers at WTP. More importantly, the results not only highlight the usefulness of this analysis for the integrity management of the floating cover but also show the value of clearly defined markers on the floating cover to facilitate the calculation of the cover's state of strain.

**Keywords:** classic shell finite assumptions; in-place movement; structural health monitoring; non-contact inspection; UAV photogrammetry; HDPE membrane; floating cover

## 1. Introduction

Melbourne Water Corporation (MW) owns and operates a large wastewater treatment facility in Werribee, Victoria, Australia, known as Western Treatment Plant (WTP). WTP is a combination of lagoon systems and activated sludge plants that treat more than 50% of Melbourne's wastewater [1]. The raw and untreated wastewater flows into the first (also known as primary) treatment lagoons at WTP to allow preliminary sedimentation of the suspended solids in the raw sewage. The first 8 hectares of these primary treatment lagoons are covered with multiple high-density polyethylene (HDPE) geomembrane sheets to capture all emissions from the lagoon and support the anaerobic processes which start the breaking down of the sewage and that produce the biogas which is converted into electricity that operates the plant, see Figure 1.

At the WTP, unfiltered raw sewage flows into the covered end of the lagoons. During anaerobic digestion of the sewage, a combination of solids, fats, oils, greases, fibrous materials, and low-density sludge can entrap the small bubbles of biogas and then be transported up through the sewage to the underside of the cover. The accumulation of this semi-solid matter creates a "scum" that can increase in volume and hardness over time, and some may even adhere to the underside of the cover and/or solidify [2]. Large accumulations of this solid matter under the cover are called scumbergs. In combination

with the wind blowing across the top surface of the cover and the continual inflow of raw sewage (hydrodynamic loading) under the cover, the scum can further deform and stress the floating cover to the extent that it may approach the intended limits of the covers' design. It is important to note that wind or hydrodynamic loading has the potential to move the cover even in the absence of scum, and this can manifest itself by observable large-scale wrinkles in the cover.

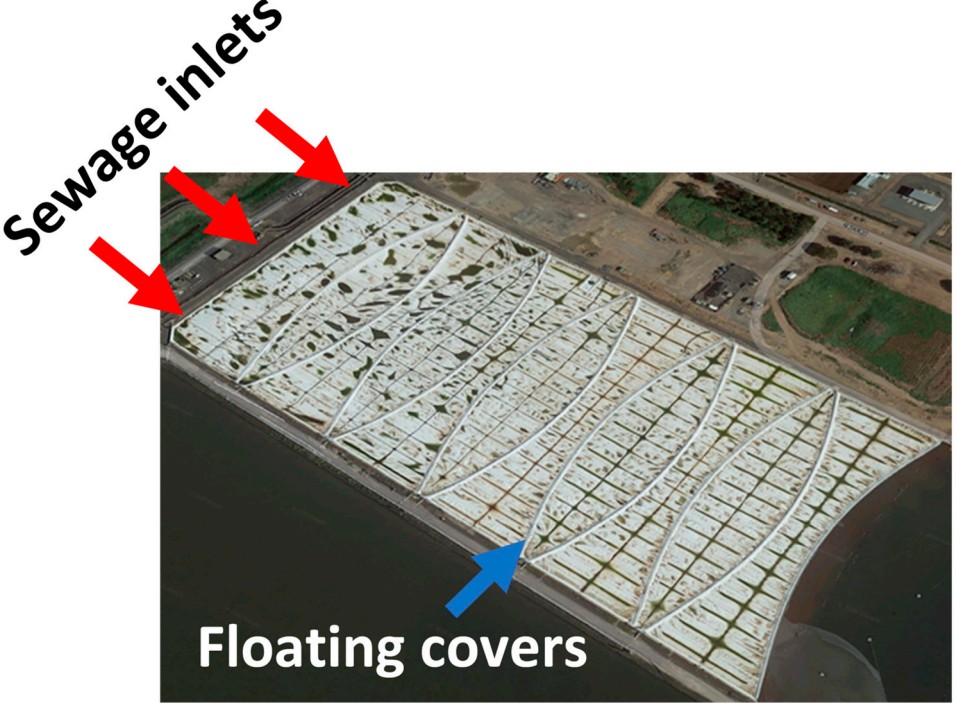

**Figure 1.** Aerial view of one of the two large (i.e., ~80,000 m$^2$) floating covers at WTP.

UAV photogrammetry (using an optical vision camera integrated UAV) is a safe and time-efficient assessment technique [3] especially for monitoring large-scale engineering structures like floating covers at WTP. With sufficient overlapping of discrete aerial images, photogrammetry can be used to establish and derive the geometric relationship between the aerial images using commercialised software (i.e., Metashape or Pix4D). The common points between overlapping images can be used by a feature matching technique to generate an orthophoto, digital surface model (DSM), and digital elevation model (DEM) of each cover. The accuracy and reliability of UAV-aided photogrammetry have been validated [4] and its potential for remoting sensing applications is also discussed in the literature [5,6].

The research and development of field scale strain measurement have received significant attention over the last decade. Most works were predominantly reported on in-plane strain measurement using the digital image correlation (DIC) techniques using two fixed cameras and require patterns to be imposed onto the structure. The DIC techniques were reported successfully measure strain 100% on a ductile thermoplastic specimen [7]. Similarly, Salvini et al. [8] successfully demonstrated large strains (up to 40%) could be measured from a polyethylene structure with a painted speckle pattern. However, it is noted that large deformation can degrade the pattern and can lead to decorrelation [9], and hence affect the accuracy of measurement.

The ability to predict the strain field from 3D information (out-of-plane) is considered very challenging. Baqersad et al. [10] successfully demonstrated the integration of 3D structural out-of-plane deformation with finite element analysis (FEA) for full-field strain prediction on the wind turbine blade. They integrated photogrammetry output describing the 3D dynamic response of a wind-turbine blade with finite element analyses to predict the dynamic strain experienced by the blade [11–13]. Their analysis is linear and strain

measurement is within 400 $\mu\varepsilon$. Ong et al. [14] constructed a finite element model by importing the 3D geometry from UAV-aided photogrammetry of PVC membrane (4.6 m $\times$ 4.6 m) as 3D triangular mesh and converted it into plate elements. As the PVC membrane was subjected to large deformation (1.4%), the finite element model was analysed using a non-linear solver to find the change in strain state between the initial state and the deformed state. Vien et al. [15] successfully demonstrate the application of non-linear finite element analysis and statistical approach to back-calculate the strain field of HDPE membranes with wrinkles without providing the boundary conditions at the edge of the membrane. Their works also compare the accuracy of the median filter on the raw DEM obtained from a deformed membrane. The works reported on the measurement of out-of-plane deformation are often associated with the need for advanced computation (i.e., self-written scripts to calculate the incremental loads in steps), computationally costly, and are sensitive to the quality of the photogrammetry output. It is noted that any noisy data in the raw DEM can make the finite element model unsolvable and over-smoothed data can significantly affect the accuracy of the strain measurement.

The work presented aims to facilitate the development of UAV-aided photogrammetry to quantify the change in the state of strain on large-scale membrane structures resulting from changes in the geometry of the membrane due to both out-of-plane and in-plane displacement. The work described in this paper demonstrates the potential of using changes in the spatial geometry of a membrane structure to ascertain its state of strain via finite element analysis. The size of the membrane called for a capability that can define the spatial geometry of a large structure.

This paper proposes to apply the finite element formulation, in a less computation costly way, to calculate the change in the lateral movement of the cover over five years. The work presented in this paper will build on the previous work as described in [16,17]. The proposed methodology is used to quantify the lateral movement and does not reflect the large deformation and non-linear wrinkling of the cover. However, this process is useful in identifying the regions of membrane stretch and wrinkles. In this respect, this analysis technique is good for quantifying the global movement of the floating cover. Distinguishable features on the floating cover are first identified and their ($x$, $y$ and $z$) coordinates are obtained from both orthophoto and DEM. All identified features on the surface of the cover are first meshed using a 3-node triangle element and a fictitious node is then added with a constant distance normal to the centre of each triangle element formed (3+1 tetrahedral element). By applying the classical shell element assumptions to a tetrahedral element, the strain tensor for each element on the floating cover is calculated. It is important to note that the "strain" calculated in this paper relates to the global deformation of the cover. In this respect, we use this formulation to enable us to determine the change in the global lateral movement of the cover.

## 2. UAV Surveillance of the Floating Cover at the WTP

A series of UAV flights have been conducted at WTP over five years to capture aerial images of the large floating covers. The details of each surveillance are summarised in Table 1. The orthophoto of the floating cover at the lagoon is shown in Figure 2. The maiden scan was conducted by an authorised remote pilot who manually piloted the UAV at approximately 60 m above ground level with the image captured at 2 s intervals. The percentage of overlapping is estimated to be 40% and 30% for side and forward overlapping. The remaining UAV surveillances were conducted using a pre-programmed third-party software, Pix4DCaputre to perform a pre-configured single grid flight path with more than 70% overlap at 50 m above the floating cover. WTP mandates a minimum flight elevation of 20 m from the surface of the covers for safety reasons.

**Table 1.** Flight parameters and Photogrammetry configurations for the UAV surveillances over the lagoon cover at WTP.

| Equipment | Fixed Wing Sensefly eBee with Sony WX220 | M600 Pro with Zenmuse X5 Camera (15 mm Lens) | | |
|---|---|---|---|---|
| Site | WTP Lagoon Cover 1 | | | |
| Mapped area | 450 m × 220 m | | | |
| Trial Scan | t1 | t2 | t3 | t4 |
| Total images taken | 133 | 793 | 928 | 929 |
| Image resolution | 4896 × 3672 | 4608 × 3456 | | |
| Flight method | Manual | Single grid programmed flight path (Pix4D Capture) | | |
| Overlap (%) | | | | |
| (I) Longer side | 30 (estimated) | 80 | 80 | 80 |
| (II) Shorter side | 40 (estimated) | 70 | 80 | 80 |
| Flight altitude | 60 m AGL | 50 m AGL | | |
| Metashape Photogrammetry setting | High Alignment & High Dense cloud configuration | | | |
| Spatial resolution for DEM (photogrammetry) | 2.75 cm/pixel | 1.14 cm/pixel | 1.11 cm/pixel | 1.11 cm/pixel |

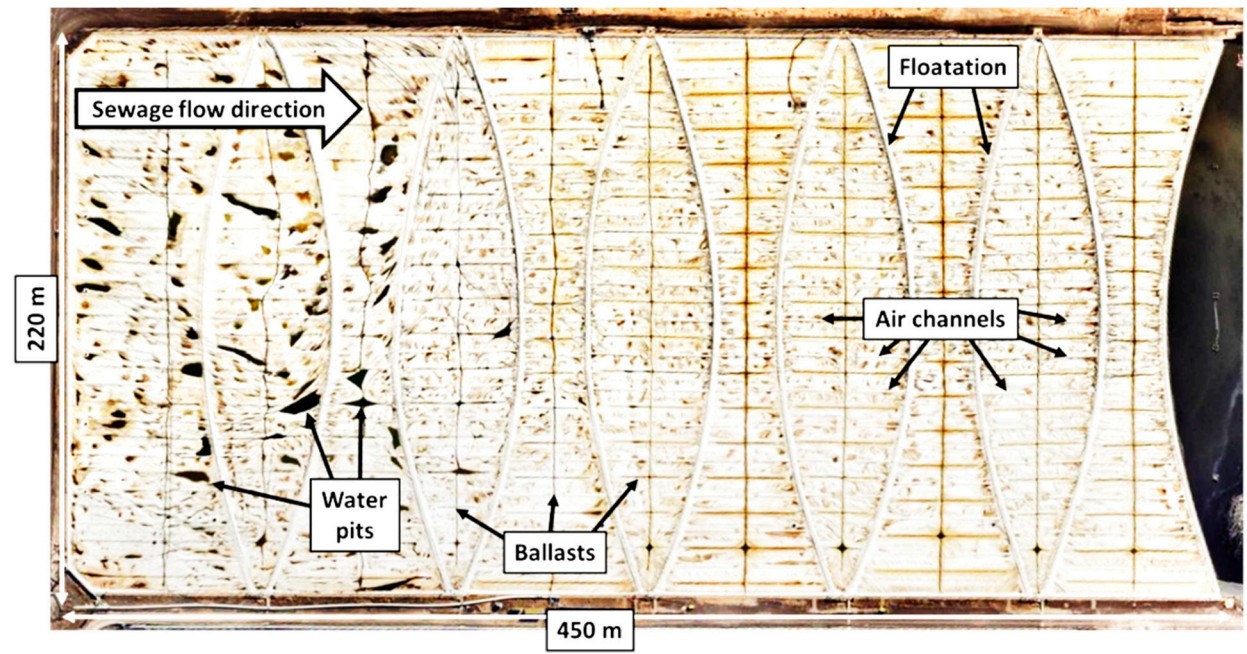

**Figure 2.** Details on Floating cover over lagoon at WTP.

Metashape Professional by Agisoft [18] was then used for post-processing of the images taken from the scan (photogrammetry) to generate a digital elevation model (DEM). The computer used for the post-processing has 2 processors (Intel® Xeon® CPU E5-2630 0@2.3GHz) with 6 cores and 128 GB of memory. Agisoft Metashape adopts computer vision algorithms as described in [19–21] which allow the user to set the quality of aligning images, building dense clouds, mesh and capable of generating a digital elevation model (DEM). All the metadata (e.g., GPS location and camera setting) of the images were first imported to Agisoft Metashape Professional for alignment purposes. Metashape allows the user to define the quality of aligning images, and building dense clouds and mesh. All settings were set to high when post-processing the images acquired from all flights. A total of 8 ground control points (GCPs) were used to improve the accuracy of the alignment. The locations of the GCPs are presented in Figure 3.

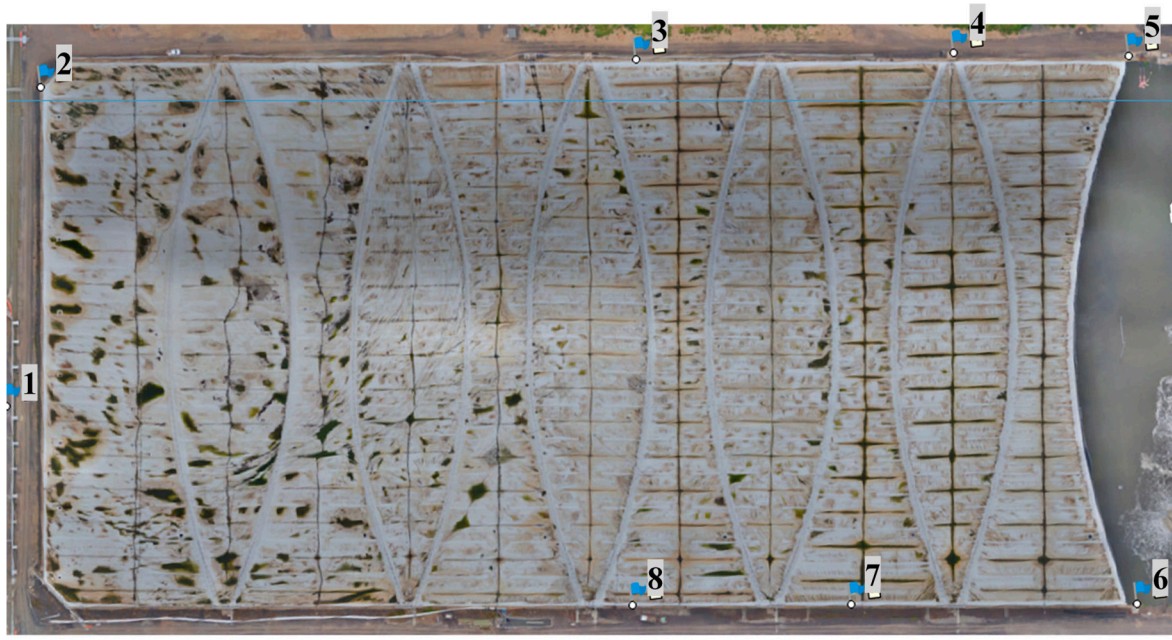

**Figure 3.** Orthophoto of the covered anaerobic lagoon (CAL) at WTP with GCPs.

UAV photogrammetry has been deployed to process the aerial images to construct the floating covers' orthophoto and DEM. Each constructed orthophoto and DEM will have a slightly different spatial resolution. Since the maiden scan was conducted at a higher altitude with a lower overlapping percentile, the spatial resolution of its DEM is expected to be lower than the scans obtained on other flights. The DEM, on its own, only describes the shape of the cover (i.e., how the cover is displaced vertically). Together with orthophoto, which allows us to determine the lateral movement of the cover part, they will contribute to a new solution that provides information on the actual movement of the cover over time, and therefore contribute to our ability to predict the strain of the cover.

## 3. In-Plane Movement Estimation Technique

For the in-plane deformation estimation for the floating cover at WTP, finite element formulations can be used. The concept of applying the classic thin shell assumptions to the tetrahedral solid elements [22,23] is proposed to calculate the field scale in-plane deformation. To convert the solid tetrahedral element for the floating cover problem, some shell assumptions have to be made to simplify the calculations, viz. (1) plane stress condition, where $\sigma_{zz} = 0$ (2) the element is not stretched in the thickness direction, where $\varepsilon_{zz} = 0$ (3) plane sections remain plane, (4) discrete Kirchhoff approach where the thickness is assumed to be normal to the surface ($\varepsilon_{xz} = 0$, $\varepsilon_{yz} = 0$) and (5) introducing a fourth fictitious node situated at the centre of the triangle and with a constant distance normal to the surface of the triangle element. Figure 4 shows the proposed element in 3D coordinates with points 1–3 (3 node triangle element) on the surface of the floating cover and point 4 as an imaginary point with a distance away normal to the centroid of the triangle formed by points 1–3. The nominal thickness of the cover is set to be 2 mm from the upper surface of the floating cover. The membrane behaviour of the element is derived from the solid volume tetrahedron element [24]. After applying the assumptions, the strain tensor of the tetrahedral element can be reduced and rewritten as Equation (1).

$$\text{Strain Tensor} = \begin{Bmatrix} \varepsilon_x \\ \varepsilon_y \\ \varepsilon_{xy} \end{Bmatrix} = \frac{1}{6V} \begin{Bmatrix} \beta_1 & 0 & \beta_2 & 0 & \beta_3 & 0 & \beta_4 & 0 \\ 0 & \gamma_1 & 0 & \gamma_2 & 0 & \gamma_3 & 0 & \gamma_4 \\ \beta_1 & \gamma_1 & \beta_2 & \gamma_2 & \beta_3 & \gamma_3 & \beta_4 & \gamma_4 \end{Bmatrix} \begin{Bmatrix} u_1 \\ v_1 \\ u_2 \\ v_2 \\ u_3 \\ v_3 \\ u_4 \\ v_4 \end{Bmatrix} \tag{1}$$

$$\begin{bmatrix} 1 & x_1 & y_1 & z_1 \\ 1 & x_2 & y_2 & z_2 \\ 1 & x_3 & y_3 & z_3 \\ 1 & x_4 & y_4 & z_4 \end{bmatrix}^{-1} = \frac{1}{6V} \begin{bmatrix} \alpha_1 & \alpha_2 & \alpha_3 & \alpha_4 \\ \beta_1 & \beta_2 & \beta_3 & \beta_4 \\ \gamma_1 & \gamma_2 & \gamma_3 & \gamma_4 \\ \delta_1 & \delta_2 & \delta_3 & \delta_4 \end{bmatrix}, \tag{2}$$

where

$$6V = \begin{vmatrix} 1 & x_1 & y_1 & z_1 \\ 1 & x_2 & y_2 & z_2 \\ 1 & x_3 & y_3 & z_3 \\ 1 & x_4 & y_4 & z_4 \end{vmatrix}, \tag{3}$$

where $\varepsilon_x$ is the strain in direction $x$, $\varepsilon_y$ is the strain in direction $y$, $\varepsilon_{xy}$ is in-plane strain, $u$, and $v$, are nodal displacements in $x$ and $y$ directions, respectively, the $\beta$ and $\gamma$ are matrix coefficients that can be obtained from the explicit inversion of Equation (2). The equivalent change and principal changes can then be calculated using Equations (4) and (5) respectively.

$$\varepsilon_{eq} = \frac{2}{3}\sqrt{\frac{3}{2}\left(\varepsilon_x^2 + \varepsilon_y^2\right) + \frac{3}{4}\left(\varepsilon_{xy}^2\right)} \tag{4}$$

$$\varepsilon_{1,2} = \left(\frac{\varepsilon_x + \varepsilon_y}{2}\right) \pm \sqrt{\left(\frac{\varepsilon_x - \varepsilon_y}{2}\right)^2 + \left(\frac{\varepsilon_{xy}}{2}\right)^2}. \tag{5}$$

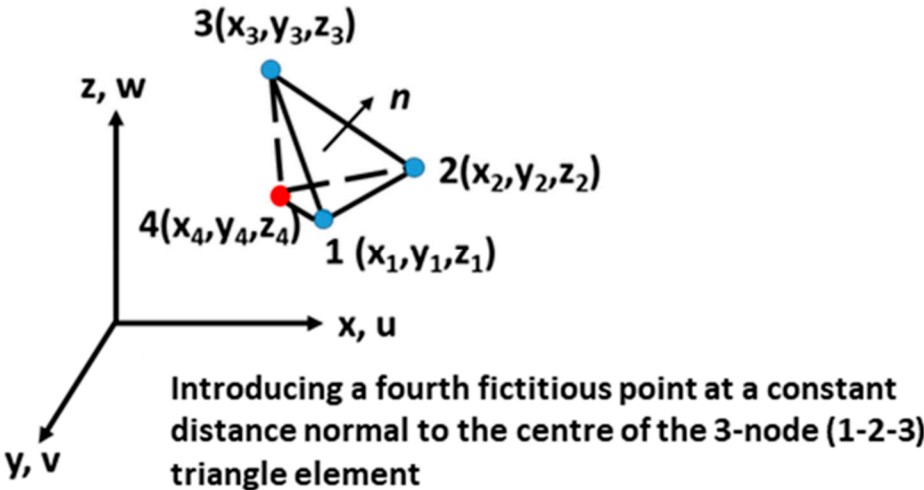

**Figure 4.** 3 + 1 node tetrahedral element.

## 4. Conceptual Validation—Finite Element Analysis

To verify the proposed analysis technique, a model of a piece of rectangular thin HDPE membrane (400 mm × 120 mm × 10 mm) was constructed using ANSYS 19.2, see Figure 5a. The Young's modulus of the HDPE is 1.1 GPa and its Poisson's ratio of 0.418 was used. A 10 mm mesh size was used to construct 480 hexahedron elements for the membrane.

Assuming the HDPE remains within the elastic range, a point load of 250 N was applied at the centre (20 mm × 20 mm) of the membrane with one end of the membrane set up with fixed and the other end set as a roller (no displacement in $z$) indicated in Figure 5b. The directional deformation and strain (in terms of $x$, $y$ and $z$) of the membrane were then calculated, see Figure 5c.

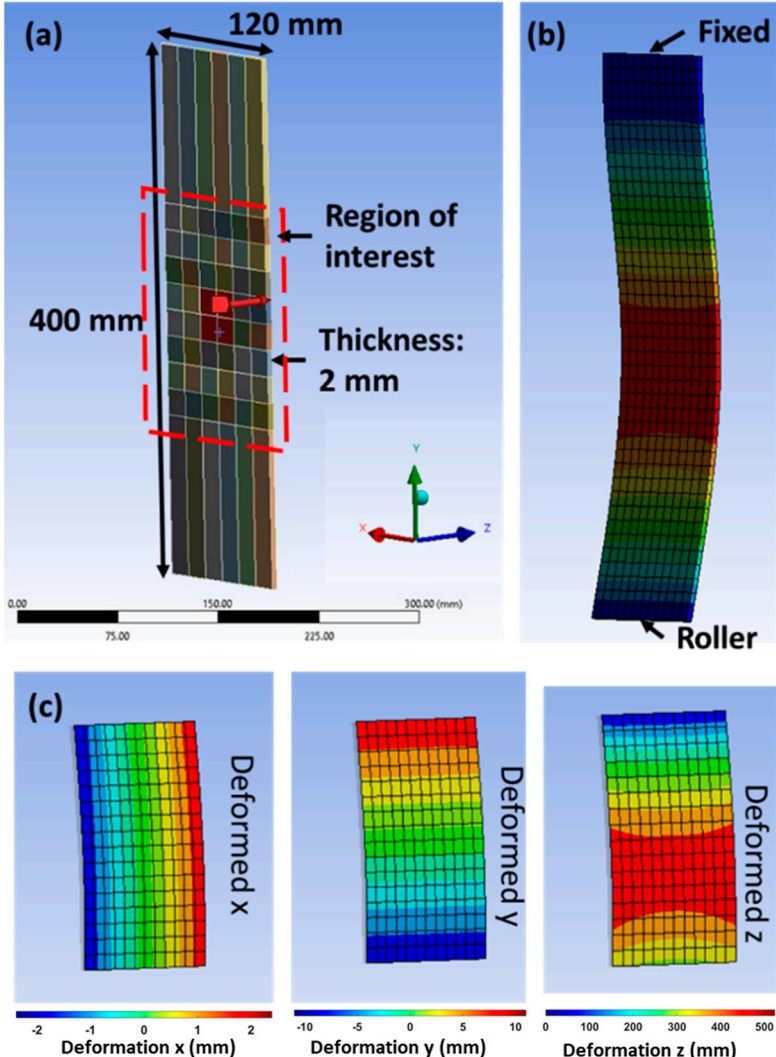

**Figure 5.** (**a**) A rectangular thin membrane model; (**b**) deformed shape after applied load at the middle of the thin membrane; (**c**) directional deformation of the membrane model.

The initial nodal coordinates and the deformed nodal coordinates within the region of interest are to simulate inputs from the photogrammetry. Therefore, only the nodal coordinates on the surface of the membrane were exported to test out the algorithm developed. The output from the algorithm described above was used to calculate the strain field.

A total of 221 nodes from the region of interest on the surface were exported and imported into MATLAB 2019a, see Figure 6a. These nodes were then meshed using the built-in function Delaunay Triangulation [25] in MATLAB to form the 3-nodes triangular elements. The fourth fictitious point is then introduced with a nominal constant distance of 2 mm from the centre of the surface of the 3-node triangle element to form a 3+1 node tetrahedral element, see Figure 6b. Since the displacement can be calculated from each node, the in-plane strain and movement can then be calculated with respect to its original coordinate using Equation (1).

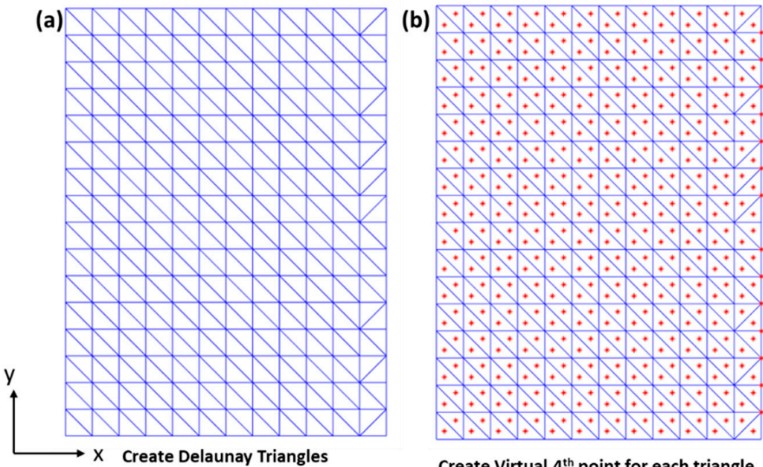

**Figure 6.** (**a**) Triangular mesh generated using Delaunay Triangular given the coordinates within the region of interest; (**b**) calculating the centroid of the generated triangle as the fourth virtual point for each triangle mesh.

Figure 7 shows good agreement between the strain fields obtained from FEA and that calculated from the algorithm presented above. Figure 8 shows the strain distribution along a given line on the membrane. The agreement attests to the veracity of the algorithm developed to predict the displacement and, hence, strain fields in the deformed membrane structure. Due to the limited number of outstanding features in the membrane structure at the WTP, we will opportunistically use their known locations in conjunction with the photogrammetry outputs to calculate the global displacement fields.

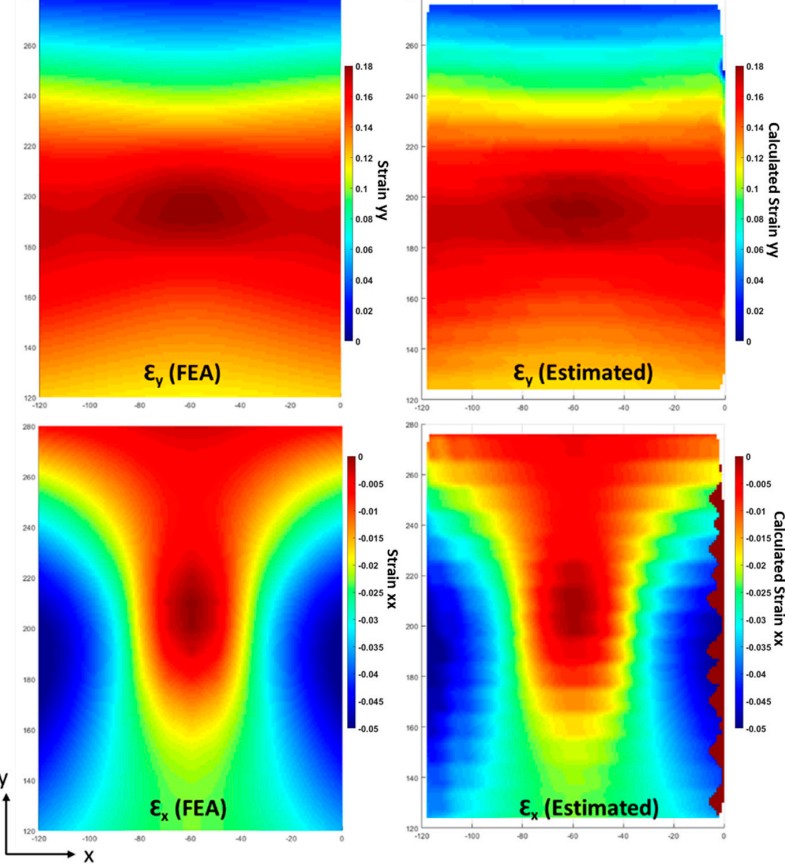

**Figure 7.** Strain contour plots along x and y for FEA and estimated using the proposed technique.

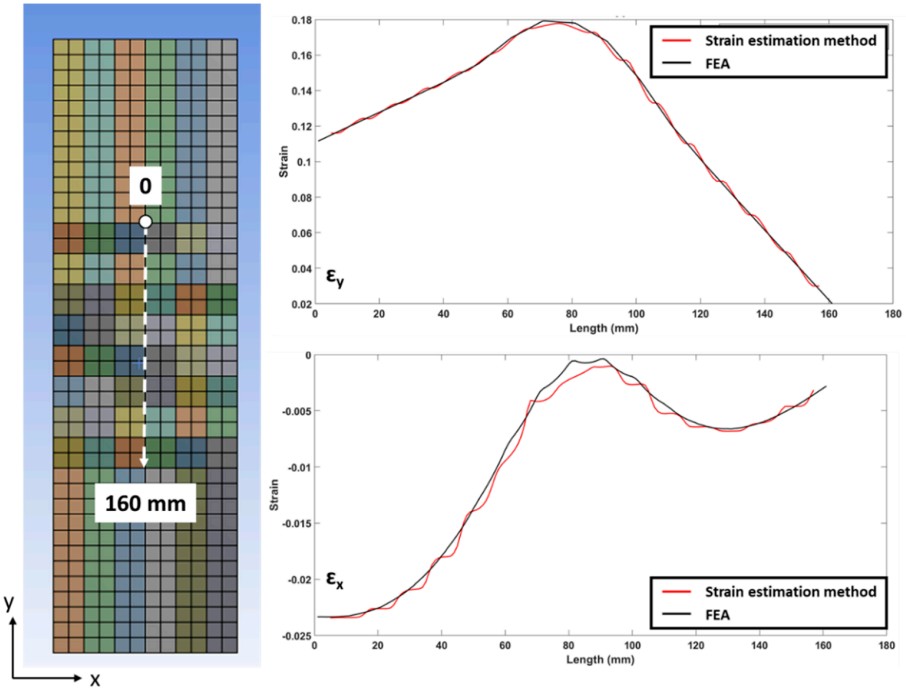

**Figure 8.** Strain comparison along the centre line of the region of interest.

## 5. Non-Contact In-Plane Movement Prediction Field Trial for WTP Floating Cover

Figure 9 shows four sets of orthophoto and DEM obtained using UAV photogrammetry as described in Section 2 and [16,17], which are imported to MATLAB 2019a for processing. The approximate time frames of these aerial scans were conducted in (1) t1 = 2 years, (2) t2 = 5 years, (3) t3 = 5.5 years and (4) t4 = 6 years after the installation of the floating cover at WTP. The DEMs of all the scans from one of the covers are presented in Figure 10. The difference in elevation (out-of-plane deformation) can be observed over time from t1 to t4. However, the deformation of the floating cover is not only restricted to the elevation (*z*-direction), but also the in-plane direction (*x* and *y*-directions). Out-of-plane and in-plane deformation of the floating covers are important as both information will provide a more complete state of deformation of the floating cover.

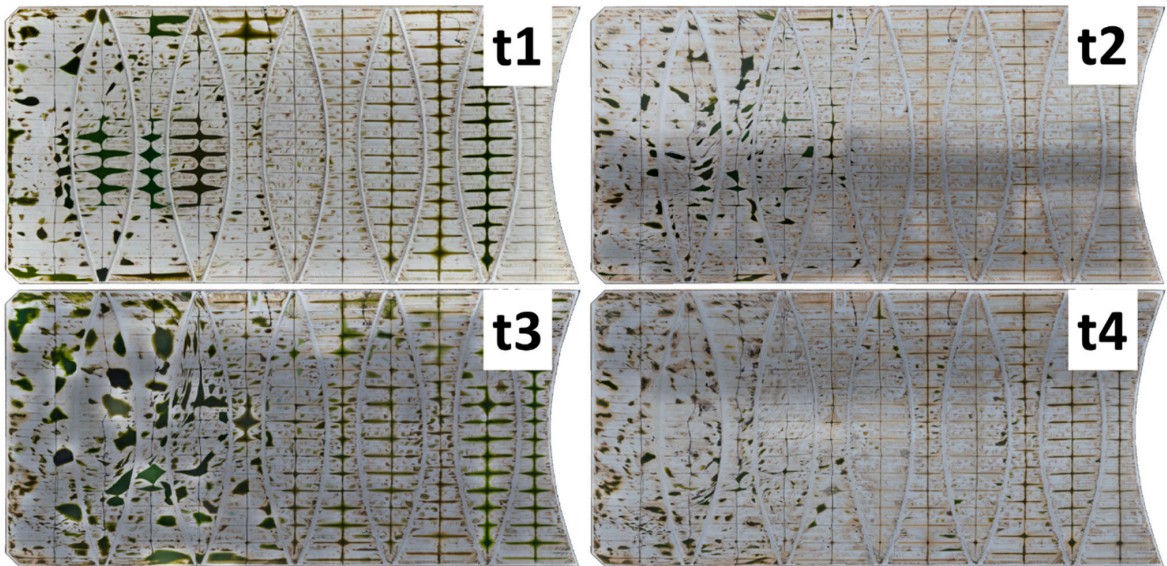

**Figure 9.** Orthophoto of floating cover from t1 to t4.

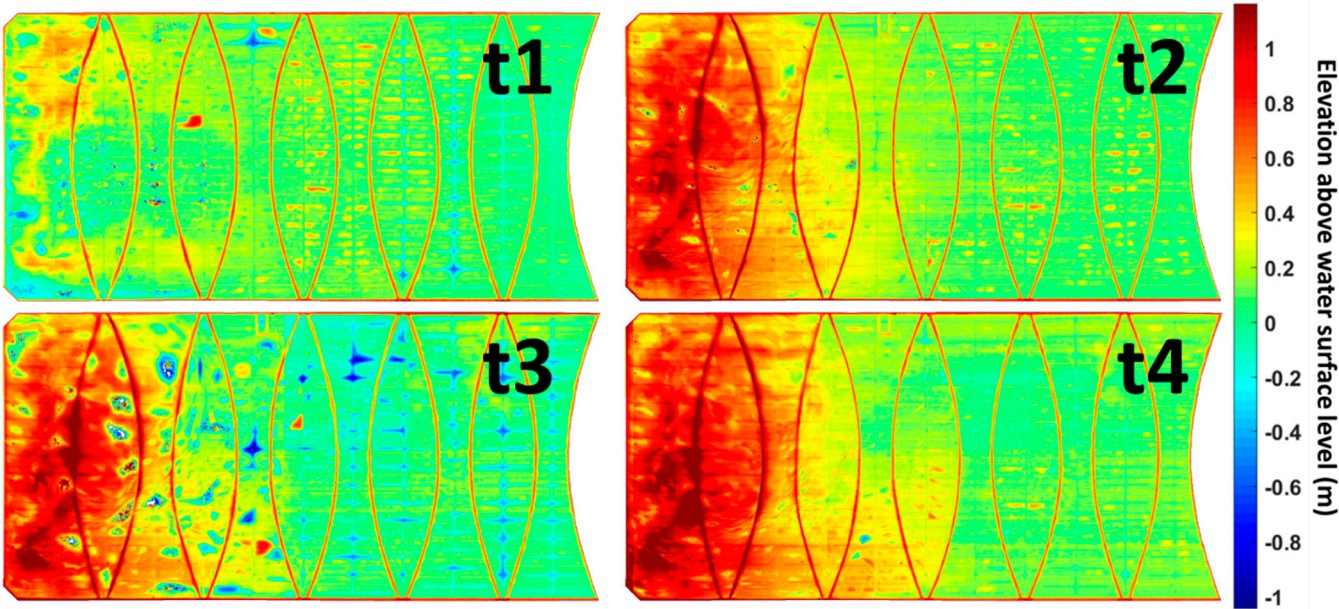

**Figure 10.** Digital elevation model of floating cover from t1 to t4.

The methodology proposed in this paper would help in quantifying the in-plane movement of the floating cover. To adopt the proposed technique, some distinguishable features on the floating cover are required. There are a few features that can be identified on the floating cover at WTP, viz. (1) end tip of secondary flotation, (2) portholes, (3) ballast retaining sleeve and protective membrane, and (4) restraining cable on the primary flotation, see Figure 11. However, the tracing work is still manually done because some of these features can easily be fully or partially obscured by dirt and water on the floating cover and can be hard to identify without high spatial resolution data.

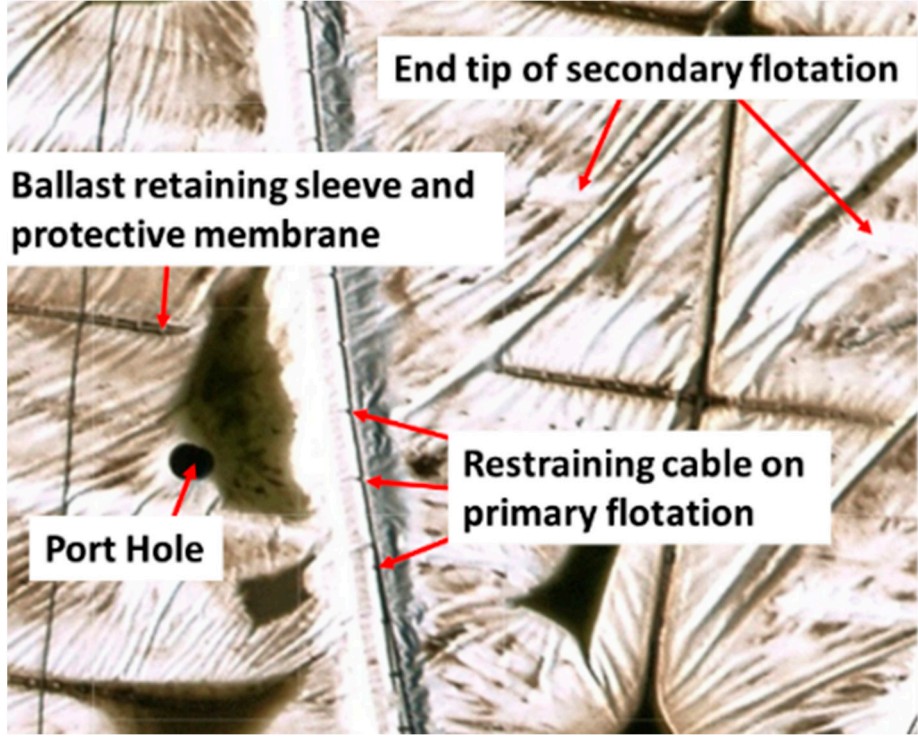

**Figure 11.** Distinguishable features on the floating cover at WTP.

A total of 792 points were successfully identified on the floating cover as indicated in Figure 12a. The elevation (*z*) of these identified features is then obtained using the corresponding DEM. A Delaunay triangulation method was again used to construct the triangular mesh in three-dimensional (3-D) space, see Figure 12b. The process was then repeated for all the scans and hence the displacement vector components (*u*, *v* and *w*) for each node can be evaluated. The in-plane movement of the cover can be calculated with respect to t1 using the strain tensor as shown in Equation (1). It is also noted that the spacing between some of the distinguishable features is more than 25 m apart and hence the calculated strain value is only used as an indicator to quantify the relative in-plane movement. Therefore, from here on, the strain tensor will be used to provide an insight into the in-plane movement which will be used to indicate the global deformation of the floating cover.

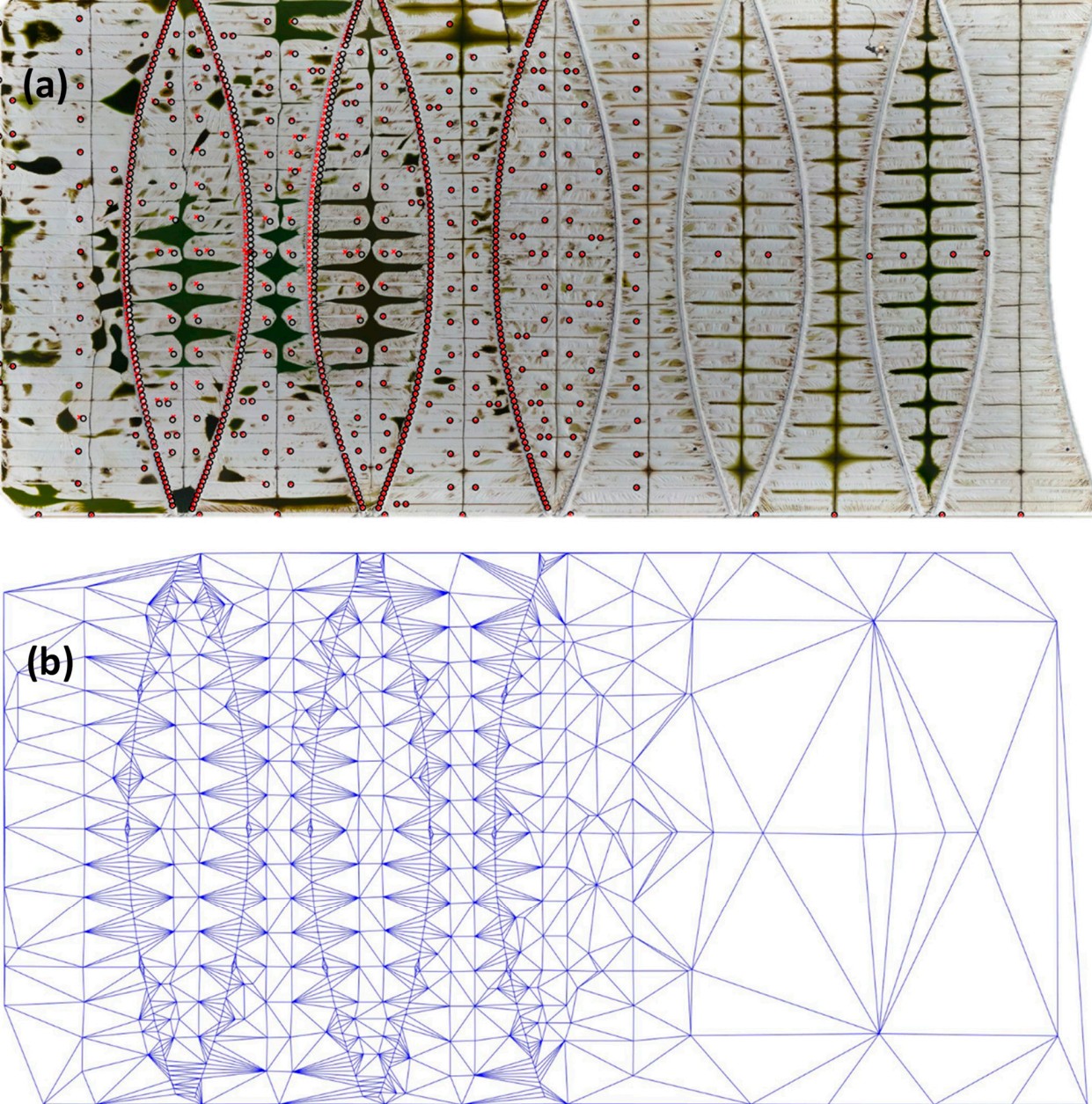

**Figure 12.** (**a**) Distinguishable features on the floating cover; (**b**) triangular mesh of the distinguishable features using Delaunay Triangulation algorithm in MATLAB.

The in-plane movements in the *x*, *y* and *xy* directions were first calculated for each scan relative to t1 using Equation (1) (i.e., the "relative" in-plane movement). The relative equivalent in-plane movement of each scan is then calculated using Equation (2) with respect to t1 to better show the movement of the cover over a longer period. These calculated "equivalent" in-plane movements are presented in Figure 13a–c. A calculation of equivalent in-plane movement is also done with respect to one scan earlier (i.e., t3 with t2 and t4 with t3) and presented in Figure 13d,e to show the in-plane movement over a shorter period. In short, Figure 13 clearly shows that the relative global deformation of the floating cover can be identified both in short term and long term. This equivalent in-plane movement will depict the regions of significant movement of the cover.

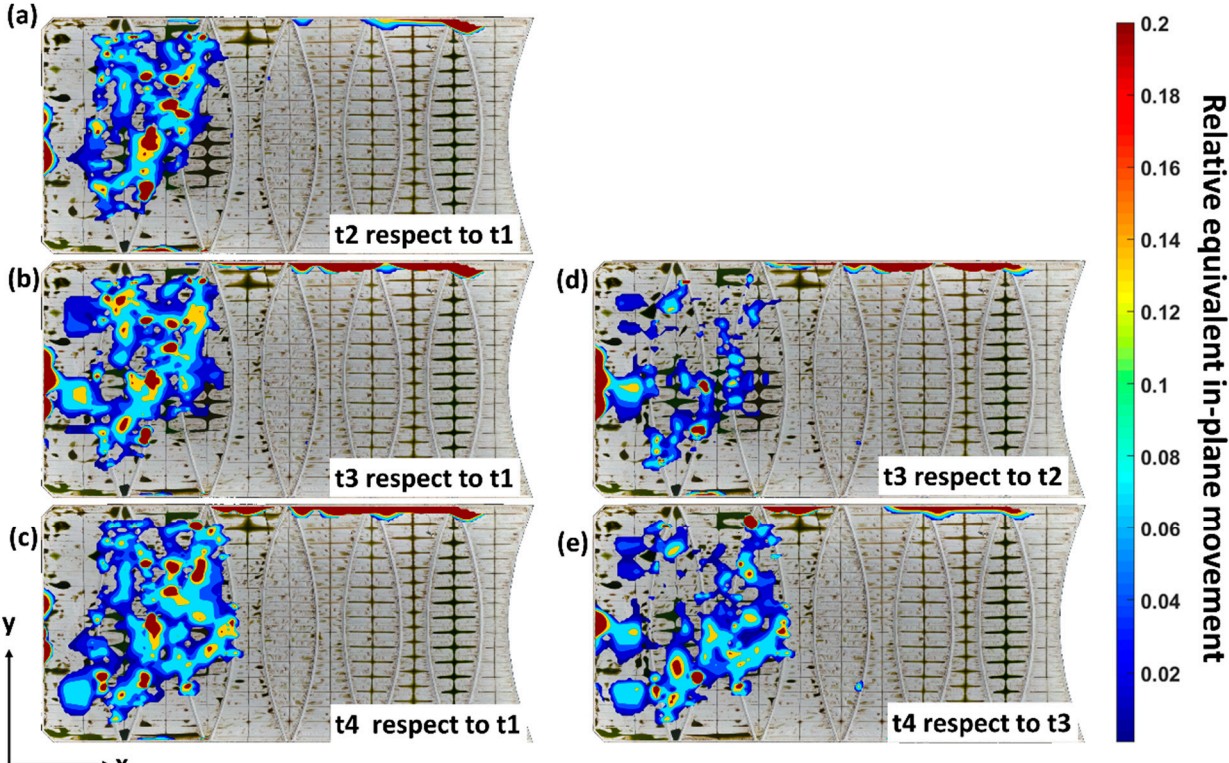

**Figure 13.** Relative equivalent in-plane movement for (**a**) t2 with respect to t1, (**b**) t3 with respect to t1, (**c**) t4 with respect to t1, (**d**) t3 with respect to t2 and (**e**) t4 with respect to t3.

To indicate the regions of tensile stretch and compression leading to cover wrinkling, the results were further processed to provide the principal values of the global displacements. To quantify the in-plane movement, the principal strain tensor was also calculated using Equation (5). We expect the negative principal values (i.e., compression) to show regions of wrinkling, whilst the positive principal values will indicate regions of membrane stretch (i.e., tension). This is because wrinkles will form on the floating cover as a result of in-plane movement. The wrinkle profile can be calculated based on the method proposed in [17] on each of the scans. The outcomes for each scan are presented in Figure 14. There were no obvious wrinkles for the scan taken at t1. By following the time-lapse of the wrinkle profiles, it is noted that the wavefront (red lines) of the wrinkles was moved towards the right of the lagoon. The movement of wrinkles indicated the growth and development of scum beneath the cover. The blue and the red regions on the left of Figure 14 denoted regions of negative principal values that indicate membrane wrinkling, and tensile stretches, respectively. These results are compared with results from [17] that show the presence and the progression of cover wrinkles. The good agreement between these results shows the potential use of this opportunistic analysis proposed that utilises the positions of a limited

number of features on the cover, the cover's digital elevation model and its orthophoto. The latter 2 quantities were obtained from photogrammetry.

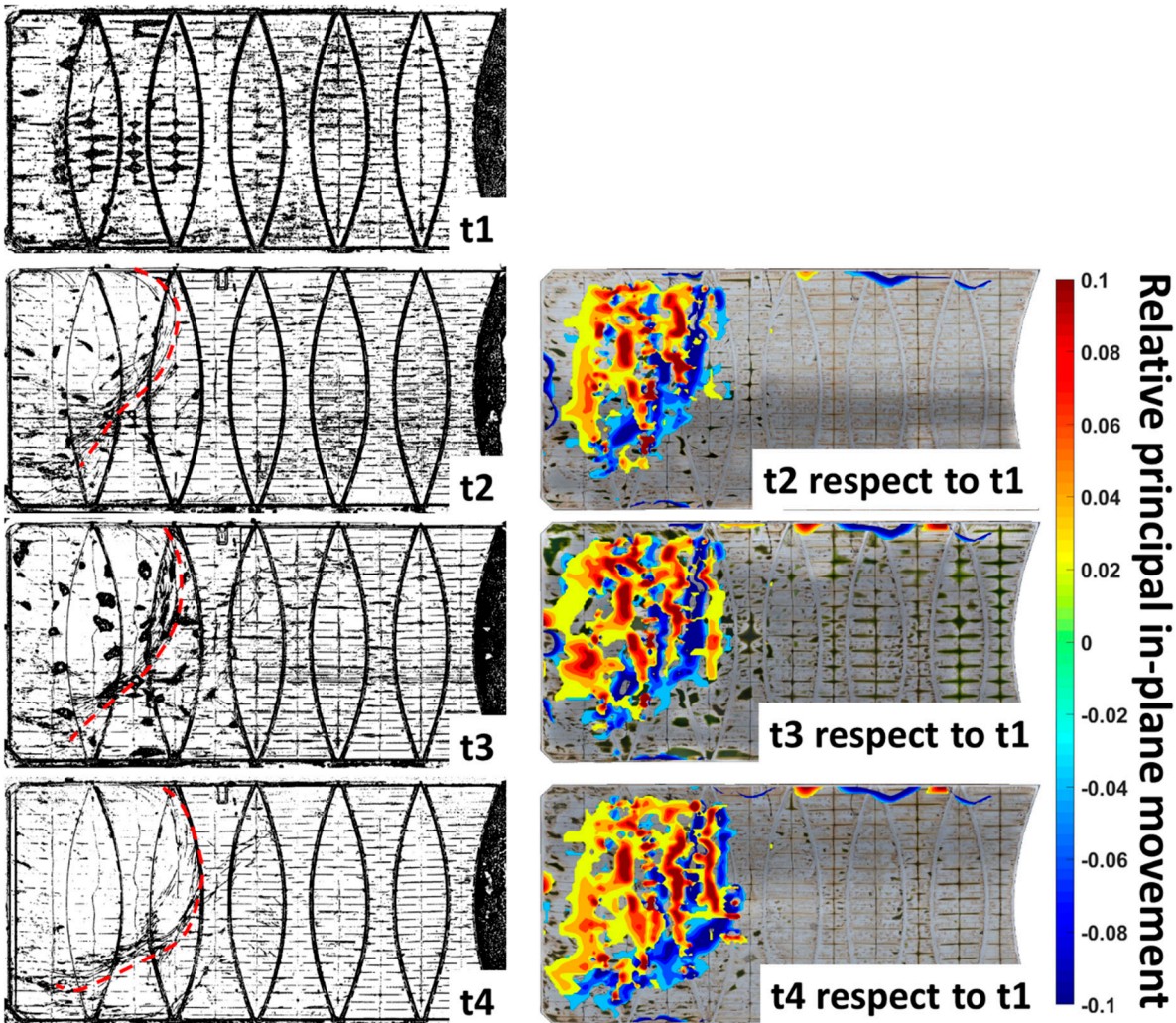

**Figure 14.** (**left column**) Wrinkles profile (wavefront—red) of floating cover from t1–t4; (**right column**) Principal in-plane movement of floating cover with respect to t1.

## 6. Practical Implications of Work Reported

This preliminary study shows that the relative in-plane movement can be calculated using the concept of the proposed methodology. This will form an important tool for the integrity management of the floating cover. Whilst the vertical displacement of the cover can be determined using the DEM, the orthophoto constructed from the images from the UAV has been shown to provide useful information to quantify the lateral cover movement. It is evident that the work reported in this paper is opportunistic as it relied on the tracking of movement of existing features on the cover. Although the paper has shown that the lateral global movement of the cover can be adequately calculated, a deliberate installation of markers on the cover with the correct spatial separation will provide more useful information in quantifying the state of strain in local regions of the cover. The ability for continual assessment of this floating cover movement is especially important to inform the operator and the designer regarding the continued performance of the engineered assets.

## 7. Conclusions

This paper shows the viability of using an efficient UAV photogrammetry inspection methodology to calculate the relative global in-plane movement of the floating covers of the anaerobic lagoons at WTP. This paper adopted the tetrahedral element with classic shell assumptions to quantify the in-plane movement of a large-scale floating cover at WTP. A global in-plane movement is determined using the 792 distinguishable features on the cover. The results show that there is visible relative global lateral movement on the floating cover at both 5 years and 6 years after installation and the proposed methodology can also be applied for short term relative global movement quantification of the floating cover. The results demonstrate the effectiveness of this technique to calculate the relative global scale in-plane deformation of a floating cover, which can provide crucial information for assessing the structural health of the floating cover and the maintenance and operation of the anaerobic lagoons.

**Author Contributions:** Conceptualization, L.W., B.S.V. and W.K.C.; methodology, L.W., B.S.V. and W.K.C.; validation, L.W. & B.S.V.; resources, T.K., J.K. and W.K.C.; formal analysis, L.W.; writing—original draft, L.W., B.S.V., D.N.B. and W.K.C.; writing—review and editing, L.W., B.S.V., T.K., D.N.B. and W.K.C.; project administration, T.K. and W.K.C.; funding acquisition, T.K., J.K. and W.K.C. All authors have read and agreed to the published version of the manuscript.

**Funding:** This research was funded by Australian Research Council Linkage Grant (ARC) LP170100108.

**Data Availability Statement:** The data presented in this study are available on reasonable request from the corresponding author.

**Acknowledgments:** This research is also financially supported by Melbourne Water Cooperation and Monash University. Their in-kind contributions are also gratefully acknowledged.

**Conflicts of Interest:** The authors declare no conflict of interest.

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
