# Peer review of "Non-Contact In-Plane Movement Estimation of Floating Covers Using Finite Element Formulation on Field-Scale DEM"

_remotesensing, doi:10.3390/rs14194761_

Round 1
Reviewer 1 Report
The paper presents an approach to calculate the lateral movement in many places of a floating cover over the raw sewage of a water treatment plant. UAV images are used to generate orthophoto, DEM and DSM. Feature matching algorithms were applied to find corresponding points in different measuring sessions and the finite element formulation is used to calculate the lateral difference.
The topic of this paper is very interesting and fits the scopes of Remote Sensing quite well. It is very interesting and new usage of UAV photogrammetry.
However, there are several critical issues which need to be improved before this paper could be considered for publication on Remote Sensing.
The detailed comments are listed as follows:
1. The motivation of this work is not clear. Why is it necessary to calculate the lateral movement of the cover over time? When compare DEM or DSM generated using UAV images from different time points, it is quite easy to know the deformation or the change in each grid. Why is this kind of change not enough?
2. The description of the process of generating orthophoto, DEM and DSM is not complete. Were there ground control points used for the georeferencing?
3. Regarding the UAV flights, how were the setting of the flight heights?
4. I do not see the reason why the finite element formulations were used. What is the connection between the UAV photogrammetry product and the movement calculated using the proposed method in the method of finite element formulations?
5. Regarding the feature detection, what kind of method was used? How was the situation of missing matching and multiple-correspondence?
6. How can you proof that the conceptual validation method can work for the evaluation?
Reviewer 2 Report
As the finite element analysis based on digital elevation models is the main innovation in the article I suggest placing it in the title of the article.
Also, add a state-of-the-art section to the paper. Show the ways and cases of FEA in other approaches.
Fig. 10: show all meshes for t1 - t4.
Fig. 11 & 12: Why changes are shown only for selected areas of the surface? What happens in the remaining parts? Show unis in the scale. Following these diagrams perform a quantitive analysis of this phenomenon, eg. calculate the field of the surface above a certain threshold or the average strength of the movement, etc.
Remember that the movement estimation is in the title so fulfill the subject.
Extend the bibliography part with other items from the state-of-the-art section. Keep the proper format of the bibliography, add a year to [5].
Reviewer 3 Report
The paper presents a very interesting approach for calculating the relative global in-plane movement of the floating covers of a wastewater lagoon with UAV photogrammetry inspection.
The following revisions are proposed:
· Fig 1 should be enlarged, use all available width so that the reader can better recognize the different characteristics of the lagoon
· Lines 68-70. Please include more recent references for the accuracy and the reliability of UAV photogrammetry
· State of the art and recent related research section should be included
· Please explain how the level of accuracy of the produced orthophoto and DEM is effecting the proposed approach
· Line 85: Since distinguishable features at the floating cover are mentioned for the first time, please describe what are these features
· Lines 162-163: Please add some specs from the UAV pictures used (number of pictures used, resolution, etc). Please also describe the process of obtaining the DEM using UAV photogrammetry
· Figure 9. The marks on the features are not clear. Better use a more intense color. A frame indicating the ROI on the left side should be included
Round 2
Reviewer 1 Report
I think that the revision is good. The current version looks more clear in terms of explanation and elaboration of methods. I am satisfied with the response to my comments except the question Nr. 5:
Regarding the feature detection, what kind of method was used? How was the situation of missing matching and multiple-correspondence?
The features here are meant as special image points that could be used for image matching. Normally, people use SIFT or SURE or others.
Did you fully use software for the 3D generation process, so that you are not aware the process in 3D point clouds generation?
In addition, there are problems with figures in the manuscript. In some figures, the legend is missing. And the results of triangulation is not necessary to be visual presented in figure.
Reviewer 2 Report
No further comments.
Author Response
N/A.